# Undiagnosed hypertension and its associated factors in India: A rural-urban contrast from the National Family Health Survey (2019-21)

Aritrik Das[1], Yukti Bhandari[1], Angad Singh[2], Jugal Kishore[3], Sonu Goel[4]*

1 Department of Community Medicine and School of Public Health, Postgraduate Institute of Medical Education and Research, Chandigarh, India, 2 Department of Biostatistics and Epidemiology, International Institute for Population Sciences Mumbai, India, 3 Department of Community Medicine, Vardhman Mahavir Medical College and Safdarjung Hospital, New Delhi, India, 4 School of Medicine, University of Limerick, Ireland

* sonu.goel@ul.ie

## Abstract

Undiagnosed hypertension extracts significant social cost, with money spent on complications already accounting for one-fifths of total health expenditure. Widespread socioeconomic disparities and inequity in health care access between rural and urban areas is expected. It is important to identify the different factors associated with undiagnosed hypertension in the working age population (15–49-years) residing in urban and rural areas, both of whom are vital to the economic development of our country. Data from the National Family Health Survey-5 (2019–21) for men and women aged 15–49 years was extracted and analysed. Operational definitions were prepared to identify known and undiagnosed hypertension. Distribution of undiagnosed hypertension according to sociodemographic, anthropometric and health-related behaviour was studied using frequencies and weighted proportions. Choropleth maps were used to depict state-wise distribution of undiagnosed hypertension. Multivariable logistic regression was used to find risk factors and protective factors for undiagnosed hypertension for men and women in rural and urban areas. The prevalence of undiagnosed hypertension was 11.7%. among men and 7.2% among women. The proportion of men with undiagnosed hypertension (66.3%) was significantly higher than the proportion of women (41.4%). Urban-rural differences were noted in various states. Education and empowerment of rural women through provision of means of socioeconomic enhancement and strengthening of community-based screening and referral under the national programme were some of the major policy implications of our findings. Future research is warranted in areas such as health insurance coverage, working away from home, owning a mobile telephone and other interventions to improve health-seeking behaviour in the rural areas.

**Data availability statement:** The data underlying the results presented in the study are available from Demographic Health Surveys https://dhsprogram.com/methodology/survey/survey-display-541.cfm as well as https://dhsprogram.com/data/dataset/India_Standard-DHS_2020.cfm?flag=0

**Funding:** The author(s) received no specific funding for this work.

**Competing interests:** The authors have declared that no competing interests exist.

## Introduction

Hypertension is the commonest cardiovascular disorder and a major public health problem across the world. Hypertension is usually essential or primary, silent, and asymptomatic [1]. It has a significant social cost, with money spent on their complications accounting for one-fifths of total health expenditure [2]. According to the World Health Organization, an estimated 1.28 billion adults aged 30–79 years worldwide have hypertension, among whom two-thirds live in low and middle-income countries [3]. Hypertension is a major cause of premature death worldwide. Only one in five adults with hypertension have it under control [3]. The rule of halves was coined in 1970s in the USA [4] and has been reported previously from surveys in India [5,6]. Pooled estimates from population representative studies from across the world, show that 51% among men and 41% among women with hypertension, were undiagnosed. The proportion of undiagnosed hypertension increased to 55% among men and 46% among women in East and South-east Asia [7].

The Global Burden of Hypertension study has highlighted that of the global burden of 212 million Disability-adjusted life years (DALYs) related to hypertension, 18% occurred in India in 2015 [8]. It is a leading risk factor for cardiovascular disease, which accounted for 23% of total deaths and 32% of adult deaths in 2010–2013 [9]. India has committed to meet the Sustainable Development Goals (SDG) target of reducing premature mortality from non-communicable diseases (NCDs) by one-third by 2030 when compared to 2015 [10]. Another nationwide study found that 42.3% of the individuals with hypertension were previously undiagnosed and the proportion was 12.4% higher in rural areas as compared to urban areas [11]. Among Indians suffering from hypertension, only one-fourth in rural areas and one-third in urban areas, are being treated while only one-tenth in rural and one-fifth in urban areas, have their blood pressure under control [6,12,13].

Since 2018, India's population has entered the phase where the working age population is larger than the dependent population, and this opportunity of demographic dividend will last until 2055 [13]. At the same time, rapid changes in dietary habits and lifestyle has resulted in an upsurge of non-communicable diseases over the past three decades [14]. The COVID-19 pandemic may have compounded the situation with more resources being diverted toward COVID management and away from non-COVID services which includes non-communicable disease management [15]. It is important to identify the factors associated with undiagnosed hypertension in this population who are vital to the economic development of our country and will also contribute to the growing burden on the health care system in the years to come. Given the heterogeneity in the sociodemographic conditions as well as the variability in primary health care delivery across states in India, it is hypothesized that there would be considerable inter-state variations in prevalence of undiagnosed hypertension. Widespread socioeconomic disparities and inequity in health care access between rural and urban areas in a state is also expected. Hence, estimates from both rural and urban areas in a state are required to identify priority areas for intervention, evaluate national programme delivery and guide policy formulation. The overall aim of this study was to determine the burden of undiagnosed

hypertension, its associated factors and its distribution across rural and urban areas among men and women aged 15–49 years in India (2019–21).

## Methods

### Study design

A cross-sectional analytical study involving secondary data from the fifth round of National Family Health Survey (NFHS) [16].

### Setting

The NFHS is a large-scale, multi-round survey conducted in a representative sample of households throughout India. It is conducted under the stewardship of the Ministry of Health and Family Welfare, Government of India. International Institute for Population Sciences, Mumbai, is the nodal agency for all the rounds of NFHS. Technical assistance for NFHS-5 was provided by the ICF International (Maryland USA) through the Demographic and Health Surveys Program, which is supported by USAID. First phase of NFHS-5 was conducted from June 17, 2019 to January 30, 2020 covering 17 states and 5 union territories and second phase was from January 2, 2020 to April 30, 2021 covering 11 states and 3 union territories. The survey covered 636 699 households, 724 115 women (15–49 years), and 101 839 men (15–54 years).

The NFHS-5 sample is a stratified two-stage sample. Villages in rural areas and census enumeration blocks in urban areas were the primary sampling units. Four questionnaires (titled household, woman, man, and biomarker) were developed in 18 local languages and data collection was done using computer assisted personal interviewing. This enabled International Institute for Population Sciences and ICF to run extensive data quality checks on the data from the field and to provide real-time feedback to field agencies and teams to help improve data quality.

Blood pressure was measured for all women and men aged between 15 and 49 years using an Omron® blood pressure monitor to determine the prevalence of hypertension. Blood pressure measurements for each respondent were taken three times with an interval of five minutes between readings.

Data was accessed and extracted for a week from 3rd-10th September 2022. The authors did not have access to any individual identifiers.

### Study population

For state-wise prevalence of undiagnosed hypertension, we included all men and women in the 15–49 years age group from NFHS-5 (2019–21). For proportion of undiagnosed hypertension among men and women with hypertension and its associated factors, we included the subpopulation with hypertension (diagnosed and undiagnosed).

### Operational definitions of undiagnosed and known hypertension

We derived known hypertension as presence of any one of the following three criteria: i) currently reported as having hypertension, ii) told by any healthcare provider that they had high blood pressure (BP) on two or more occasions, or iii) currently taking medication to lower BP. We derived undiagnosed hypertension if the respondents fulfilled all the following four: i) average of second and third systolic BP reading ≥ 140 and/or diastolic BP reading ≥ 90 mm Hg, ii) not reported as currently having hypertension, iii) not told by any healthcare provider previously to have had high BP, and iv) reported as not taking any medication to lower BP. A cut-off of 140/90 mm of Hg is used to diagnose and treat hypertension in India, and therefore this cut-off was used in this study.

### Data variables

We extracted the variables required to classify the status of hypertension (known/undiagnosed). In addition, socio-demographic characteristics, anthropometric data and data on health-related behaviours was extracted.

## Analysis

**Software.** Database was constructed, cleaned with Microsoft Excel (Microsoft, Redmond, WA, USA) and imported to STATA (version 16, copyright 1985–2011 Stata Corp LP USA) and Datawrapper developed by Datawrapper GmbH for analysis [17].

**Statistics.** We depicted the prevalence of undiagnosed hypertension using percentages and 95% CI. Descriptive analysis of sociodemographic, anthropometric and health-related behaviour variables was expressed for men and women residing in rural and urban areas with undiagnosed hypertension separately. Log binomial regression model has been applied to determine the factors associated with undiagnosed hypertension among all people with hypertension. The associations were summarized using adjusted odds ratios and 95% CI. Owing to the large sample size, a cut-off for public health significance was set for the adjusted odds ratios [aOR ≤ 0.8 & ≥ 1.3]. Choropleth figures have been used to depict state-wise proportion of undiagnosed hypertension in rural and urban areas among all men and women with hypertension. All the analysis has been weighted for the multistage sampling design and hence, weighed estimates have been provided.

## Ethical statement

The study was ethically approved by the Institute's Ethical Committee, Postgraduate Institute of Medical Education and Research (PGIMER), Chandigarh (IEC-08/2022–2535 dated 17.08.2022).

## Results

### Burden of undiagnosed hypertension

Out of a total of 1,01,839 men, the prevalence of undiagnosed hypertension was 11.7%. The proportion of men with undiagnosed hypertension among all men with hypertension was 66.3% (Fig 1). The proportion was similar between rural (66.2%) and urban (66.5%) areas.

Table 1 describes the distribution of men with undiagnosed hypertension according to various socio-demographic, anthropometric and health related behaviour variables.

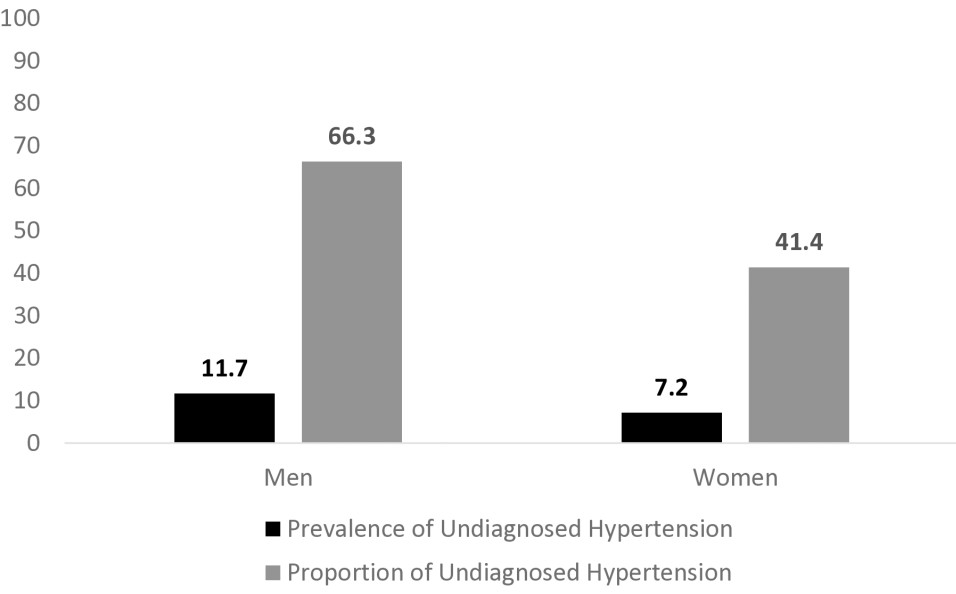

**Fig 1. Burden of Undiagnosed Hypertension among men and women aged 15-49 years from NFHS 5 (2019-2021).**

**Table 1. Proportional distribution (with 95% confidence interval) of men with undiagnosed hypertension according to various socio-demographic, anthropometric and health related behaviour variables among all men with hypertension in the 15-49 years age group in India NFHS-5, 2019-21.**

| Variables | Proportion of men with undiagnosed hypertension (95% CI) | | | Total (N) |
|---|---|---|---|---|
| | Total | Rural | Urban | |
| **Total** | **66.3 (65-67.6)** | **66.2 (64.8-67.6)** | **66.5 (63.7-69.1)** | **17936** |
| **Age group** | | | | |
| 15-19 | 66.9 (60.6-72.7) | 62.2 (55.8-68.3) | 75.9 (63-85.3) | 895 |
| 20-29 | 69.9 (67-72.5) | 68.9 (65.9-71.7) | 71.7 (65.7-77) | 3659 |
| 30-39 | 69.2 (66.8-71.5) | 69.3 (66.9-71.5) | 69.1 (63.9-73.8) | 6090 |
| 40-49 | 62.3 (60.2-64.3) | 63 (60.7-65.2) | 61.1 (56.9-65) | 7292 |
| **Cast/Tribe** | | | | |
| Scheduled caste | 68.4 (65.4-71.3) | 67.1 (63.9-70.1) | 70.8 (64.3-76.6) | 3418 |
| Scheduled tribe | 73.5 (70.3-76.5) | 75 (71.8-78) | 66.9 (56.2-76.1) | 3662 |
| OBC | 66.4 (64.3-68.4) | 66.8 (64.7-68.8) | 65.7 (61.5-69.6) | 6614 |
| Others | 62.3 (59.5-64.9) | 60.5 (57.4-63.5) | 64.7 (59.7-69.3) | 4242 |
| **Religion** | | | | |
| Hindu | 67.5 (66-68.9) | 67.6 (66.1-69.1) | 67.2 (64-70.3) | 13419 |
| Muslim | 58.4 (54.5-62.2) | 54 (49.2-58.8) | 63.3 (56.9-69.3) | 1748 |
| Christian | 69.2 (63.2-74.6) | 70.4 (63.2-76.8) | 67.1 (56.4-76.4) | 1516 |
| Others | 66 (59.7-71.8) | 67.5 (60.5-73.8) | 63 (50.3-74.2) | 1253 |
| **Marital Status** | | | | |
| Never in union | 70.3 (67.3-73.1) | 69.2 (66.2-72.1) | 71.8 (66-76.9) | 3704 |
| Currently married | 65.2 (63.7-66.7) | 65.3 (63.7-66.9) | 65.1 (62-68.2) | 13898 |
| Widowed/Divorced/Separated | 69.5 (60-77.6) | 75.5 (64.1-84.1) | 52 (35.4-68.2) | 334 |
| **Education** | | | | |
| No education/Pre-primary education | 67.8 (64.3-71.2) | 66.4 (62.6-70.1) | 72.6 (64.2-79.7) | 2125 |
| Primary | 67.1 (63.4-70.6) | 67 (63.1-70.8) | 67.2 (58.5-74.9) | 2214 |
| Secondary Education | 66.7 (64.9-68.4) | 66.8 (64.9-68.6) | 66.5 (62.7-70.1) | 10273 |
| Higher Education | 63.8 (60.5-66.9) | 62.9 (59.2-66.5) | 64.5 (59.1-69.5) | 3324 |
| **Occupation** | | | | |
| No Occupation | 67.8 (63.3-71.9) | 64.2 (59.5-68.7) | 72.6 (64.1-79.8) | 1729 |
| Professional/ Clerical | 62.7 (57.7-67.4) | 66.7 (61.1-71.8) | 60.1 (52.9-66.9) | 1514 |
| Sales | 63.9 (59.4-68.2) | 60.2 (54.6-65.6) | 66.9 (60.2-72.9) | 1801 |
| Services/household and domestic | 67.5 (62-72.6) | 70.5 (64.4-76) | 65 (56.2-73) | 1454 |
| Agricultural | 67 (64.9-69) | 67 (64.8-69.1) | 67.2 (57.5-75.7) | 5880 |
| Skilled, unskilled manual and Others | 66.7 (64.3-69) | 66.2 (63.6-68.6) | 67.4 (63-71.5) | 5558 |
| **Wealth Index** | | | | |
| Poorest | 63.4 (60.3-66.5) | 63.7 (60.5-66.7) | 59.1 (39.7-76.1) | 2981 |
| Poorer | 67.7 (65-70.3) | 67.5 (64.8-70.1) | 68.7 (58.3-77.6) | 3723 |
| Middle | 66.7 (64-69.3) | 66 (63.2-68.7) | 68.4 (62.1-74.2) | 3906 |
| Richer | 67.1 (64.2-69.9) | 68.2 (65-71.3) | 65.8 (60.7-70.6) | 3773 |
| Richest | 65.7 (62.3-68.9) | 64.9 (60.1-69.3) | 66 (61.7-70.1) | 3553 |
| **Work at home or away** | | | | |
| At home | 67.6 (66.2-69) | 67.6 (66.1-69.1) | 67.6 (64.7-70.4) | 15328 |
| Away | 58.1 (54.5-61.6) | 58.6 (55.1-62) | 56.6 (47.6-65.2) | 2608 |

*(Continued)*

**Table 1.** (Continued)

| Variables | Proportion of men with undiagnosed hypertension (95% CI) | | | Total (N) |
|---|---|---|---|---|
| **Region** | | | | |
| North | 66.6 (64.8-68.4) | 66.6 (64.3-68.8) | 66.6 (63.5-69.5) | 3743 |
| Central | 71.3 (69.5-73.1) | 71.2 (69.2-73) | 71.8 (67.7-75.5) | 3881 |
| East | 51.3 (48.1-54.5) | 54.3 (50.8-57.7) | 43.3 (36.7-50.2) | 2338 |
| Northeast | 55 (52.2-57.8) | 54.8 (51.6-58) | 55.7 (50-61.3) | 3167 |
| West | 74.8 (71.1-78.3) | 74.9 (71.2-78.3) | 74.8 (67.6-80.8) | 1614 |
| South | 71.2 (68.9-73.4) | 71.6 (69-74.1) | 70.6 (66.4-74.5) | 3193 |
| **Use of Internet** | | | | |
| Never | 67.3 (65.5-69.1) | 67.4 (65.5-69.3) | 67 (62.4-71.4) | 8032 |
| Yes | 66 (64-68) | 65.4 (63.2-67.5) | 66.6 (63.1-70) | 9427 |
| **Owns a mobile telephone** | | | | |
| Yes | 66 (61.4-70.4) | 66.1 (61.1-70.8) | 65.6 (52.8-76.4) | 1364 |
| No | 66.7 (65.2-68.1) | 66.6 (65.1-68) | 66.8 (63.9-69.6) | 16095 |
| **Usage of any type of tobacco** | | | | |
| No | 68.2 (66.4-69.9) | 67.5 (65.6-69.4) | 69.2 (65.9-72.4) | 9286 |
| Yes | 64.1 (62.1-66.1) | 64.9 (62.9-66.9) | 62.4 (57.8-66.8) | 8650 |
| **Current alcohol usage** | | | | |
| No | 66.1 (64.5-67.7) | 65.8 (64.1-67.5) | 66.7 (63.4-69.8) | 11531 |
| Less than once a week | 64.4 (60.3-68.4) | 63.9 (59.7-67.8) | 65.3 (56.4-73.3) | 2274 |
| Once a week | 69.7 (66.4-72.9) | 70.1 (66.6-73.4) | 69.1 (62.2-75.2) | 2924 |
| Everyday | 63.9 (57.6-69.7) | 66.2 (60.7-71.3) | 59.9 (45.8-72.5) | 1207 |
| **Covered by Health Insurance** | | | | |
| No | 66 (64.2-67.7) | 65.2 (63.3-67) | 67.2 (63.7-70.5) | 10977 |
| Yes | 66.9 (64.8-68.9) | 67.8 (65.7-69.9) | 65.1 (60.6-69.3) | 6959 |
| **Type of health facility recently used** | | | | |
| None | 68.5 (66.8-70.1) | 68.6 (66.9-70.2) | 68.4 (64.9-71.6) | 12261 |
| Public facility | 63.1 (60-66.1) | 61.5 (58.3-64.6) | 66.1 (59.5-72) | 3513 |
| Private facility | 61 (57.2-64.6) | 62.5 (58.5-66.4) | 58.7 (51.4-65.6) | 1999 |
| Other | 59.8 (46.4-71.8) | 57.2 (42.1-71) | 69.3 (42.4-87.4) | 163 |
| **Anaemia** | | | | |
| Severe | 50 (31.1-69) | 48.8 (29.5-68.5) | 80.7 (27.3-97.9) | 50 |
| Moderate | 59.4 (52.7-65.7) | 60 (52.3-67.2) | 57.6 (44.1-70.1) | 600 |
| Mild | 62.9 (59.5-66.2) | 63.7 (60.1-67.1) | 61 (53.1-68.4) | 2709 |
| Not anaemic | 68.3 (66.8-69.8) | 68.2 (66.6-69.7) | 68.6 (65.4-71.5) | 13659 |
| **BMI** | | | | |
| Underweight | 68.4 (63.8-72.7) | 67 (62.4-71.3) | 72.7 (60.4-82.2) | 1296 |
| Normal weight | 66.6 (64.9-68.4) | 66.5 (64.6-68.3) | 67 (63.1-70.6) | 9871 |
| Overweight | 68.8 (66.5-71) | 67.9 (65.3-70.4) | 70 (65.8-73.8) | 5290 |
| Obese | 57.8 (51.7-63.7) | 60.3 (53.9-66.3) | 55.6 (45.6-65.1) | 1318 |
| **Waist to Hip Ratio** | | | | |
| Less or equal to 0.9 for men | 67.6 (65.7-69.5) | 66.6 (64.5-68.6) | 69.5 (65.4-73.4) | 7886 |
| (Cut offs > 0.9 for men) | 65.9 (64.1-67.7) | 66.4 (64.5-68.3) | 65.2 (61.5-68.7) | 9917 |

Out of 7,24,115 women, the prevalence of undiagnosed hypertension was 7.2%. The proportion of undiagnosed hypertension among all women with hypertension was 41.4% (Figure 1). The proportion was also similar between rural (41.3%) and urban areas (41.8%). Table 2 describes the distribution of women with undiagnosed hypertension according to various socio-demographic, anthropometric and health related behaviour variables.

## Rural-urban differences in distribution of undiagnosed hypertension among men

The proportion of undiagnosed hypertension was higher among the 15–19 year (75.9% vs 62.2%) group in the urban areas as compared to the rural areas. Proportion of undiagnosed hypertension among Muslim men was lower (58.4%), especially in the rural areas (54%). The proportion of men belonging to scheduled tribes with undiagnosed hypertension was higher in rural areas than in urban areas (75% vs 66.9%). The proportion of undiagnosed hypertension was higher among men from rural areas who had been widowed/separated/divorced as compared to urban areas (75.5% vs 52%). A higher proportion of undiagnosed hypertension was found among men who were uneducated (72.6% vs 66.4%) or unemployed (72.6% vs 64.2%) in the urban areas as compared to the rural areas. In the East region, proportion of undiagnosed hypertension was higher among men from rural areas (54.3% vs 43.3%). It was higher among men who haven't used any type of health facility in recent times (68.5%). The proportion was higher among men who had used a public facility in urban areas as compared to the rural areas (66.1% vs 61.5%).

## Rural-urban differences in distribution of undiagnosed hypertension among women

Table 2 also depicts the rural-urban differences in proportion of undiagnosed hypertension among women. The proportion of women with undiagnosed hypertension was higher across all age groups (67.9% vs 37.5% in the 20–29 year age group) in rural areas as compared to urban areas,. The proportion of undiagnosed hypertension was lower among women belonging to the richest wealth index in the rural areas as compared to the urban areas (35.6% vs 40.5%). The proportion was higher among women using any type of tobacco (46.1% vs 40.3%) or alcohol (65.2% vs 54.2%) in the rural areas than in the urban areas. There was a higher proportion of undiagnosed hypertension among unemployed women in the urban areas as compared to the rural areas (42.4% vs 38.4%).

## State-wise distribution of undiagnosed hypertension among men

State-wise distribution of proportion of men with undiagnosed hypertension among all men with hypertension is visualized in Fig 1. Chhattisgarh reported the highest proportion of 83.2% (Rural-86.9%; Urban-70.3%), closely followed by Madhya Pradesh with 81.7% (Rural-82.5%; Urban-79.8%). As many as 13 states & Union Territories reported greater than 70% proportion of undiagnosed hypertension among males with hypertension in the 15–49 year age group. Several states show wide rural-urban divides. The widest urban-rural divides are seen in NCT of Delhi (Rural-93.7%; Urban-70.3%), Puducherry (Rural-88.4%; Urban-55.5%), Nagaland (Rural-70.9%; Urban-48%), Bihar (Rural-45.9%; Urban-29.7%), Andaman and Nicobar islands (Rural-68.6%; Urban-42%)and Goa (Rural-54.5%; Urban-68.4%). States with the lowest proportion of men with undiagnosed hypertension are Ladakh, Bihar, Tripura, and Jammu & Kashmir.

## State-wise distribution of undiagnosed hypertension among women

State-wise distribution of proportion of women with undiagnosed hypertension among all women with hypertension is visualized in Fig 2. Chhattisgarh reported the highest proportion of 64.3% (Rural-67.7%; Urban-54.5%) followed by Karnataka with 56.2% (Rural-56.3%; Urban-56.1%) and Lakshadweep with 56.2% (Rural-42.8%; Urban-59.1%). Maharashtra, Telengana, Gujarat and Arunachal Pradesh reported greater than 50% proportion of undiagnosed hypertension among women with hypertension in the 15–49 year age group (Fig 3). The rural-urban divide is also prominent in this distribution as seen in states

**Table 2. Proportional distribution (with 95% confidence interval) of women with undiagnosed hypertension according to various socio-demographic, anthropometric and health related behaviour variables among all women with hypertension in the 15-49 years age group in India NFHS-5, 2019-21.**

| Variables | Proportion of women with undiagnosed hypertension (95% CI) | | | Total |
|---|---|---|---|---|
| | Total | Rural | Urban | |
| **Total** | **41.4 (41.1-41.8)** | **41.3 (40.9-41.7)** | **41.8 (41-42.5)** | **126564** |
| **Age group** | | | | |
| 15-19 | 42 (40.4-43.6) | 59.6 (57.9-61.3) | 46.3 (42.6-50.1) | 6584 |
| 20-29 | 33.7 (32.9-34.5) | 67.9 (67-68.7) | 37.5 (35.8-39.3) | 26562 |
| 30-39 | 43 (42.3-43.7) | 57.7 (57-58.5) | 44.5 (43.1-45.9) | 40204 |
| 40-49 | 43.9 (43.3-44.5) | 54.5 (53.8-55.1) | 41.1 (40-42.3) | 53214 |
| **Cast/Tribe** | | | | |
| Scheduled caste | 39.9 (39-40.7) | 39.6 (38.7-40.5) | 40.4 (38.7-42.2) | 24569 |
| Scheduled tribe | 55.9 (54.7-57.1) | 57 (55.8-58.3) | 50.2 (46.5-53.9) | 22796 |
| OBC | 41.2 (40.6-41.7) | 40.8 (40.2-41.4) | 42 (40.9-43.1) | 46997 |
| Others | 38.7 (37.9-39.5) | 37 (36.1-37.9) | 41.2 (39.8-42.6) | 32202 |
| **Religion** | | | | |
| Hindu | 42 (41.5-42.4) | 42.2 (41.7-42.7) | 41.5 (40.6-42.4) | 92359 |
| Muslim | 39.9 (38.8-41) | 36.8 (35.5-38.1) | 44.2 (42.3-46.1) | 15844 |
| Christian | 43.8 (41.4-46.2) | 45.4 (42.7-48.1) | 41.1 (36.7-45.7) | 9451 |
| Others | 34.3 (32.7-36) | 33.8 (32.1-35.4) | 35.6 (31.8-39.7) | 8910 |
| **Marital Status** | | | | |
| Never in union | 46.7 (45.4-47.9) | 46 (44.6-47.4) | 47.9 (45.4-50.3) | 12236 |
| Currently married | 40.6 (40.2-41) | 40.5 (40-40.9) | 40.9 (40.1-41.8) | 105922 |
| Widowed/Divorced/Separated | 45 (43.5-46.6) | 46.1 (44.4-47.8) | 43.2 (40.4-46.1) | 8406 |
| **Education** | | | | |
| No education/Pre-primary education | 45.2 (44.6-45.9) | 45.7 (45-46.4) | 43.3 (41.6-45.1) | 38628 |
| Primary | 42.8 (41.8-43.9) | 42.5 (41.4-43.6) | 43.7 (41.4-46) | 18254 |
| Secondary Education | 39.5 (38.9-40) | 38.2 (37.5-38.8) | 41.6 (40.4-42.7) | 56146 |
| Higher Education | 37.3 (36.1-38.5) | 34.3 (32.8-35.9) | 39.5 (37.8-41.3) | 13536 |
| **Occupation** | | | | |
| No Occupation | 39.8 (38.6-41.1) | 38.4 (37-39.7) | 42.4 (40-44.9) | 12344 |
| Professional/ Clarical | 45.5 (39.1-52) | 46.4 (38.9-54) | 44.8 (35.2-54.8) | 625 |
| Sales | 46.6 (38.4-55.1) | 46 (37.5-54.6) | 47.2 (33.7-61.2) | 456 |
| Services/household and domestic | 47.4 (41.9-53.1) | 48.1 (41-55.2) | 46.9 (38.7-55.4) | 769 |
| Agricultural | 51 (48.6-53.3) | 51 (48.6-53.4) | 50.8 (40.6-61) | 3254 |
| Skilled, unskilled manual and Others | 42.8 (39.3-46.3) | 43.5 (39.4-47.7) | 41.6 (35.5-48) | 1543 |
| **Wealth Index** | | | | |
| Poorest | 43.9 (43-44.7) | 43.9 (43.1-44.8) | 42.7 (37.9-47.7) | 23203 |
| Poorer | 41.8 (41-42.6) | 41.4 (40.6-42.3) | 44.3 (41.4-47.3) | 26910 |
| Middle | 41.8 (41-42.6) | 41.4 (40.5-42.3) | 42.9 (41-44.9) | 26884 |
| Richer | 41.4 (40.5-42.2) | 40.6 (39.5-41.6) | 42.3 (40.9-43.7) | 25718 |
| Richest | 39 (38.1-39.9) | 35.6 (34.4-36.8) | 40.5 (39.4-41.7) | 23849 |
| **Work at home or away** | | | | |
| At home | 43 (42-44.1) | 42.6 (41.4-43.8) | 43.9 (41.7-46) | 17506 |
| Away | 34.9 (31.4-38.5) | 33.8 (30.3-37.5) | 36.8 (29.7-44.5) | 1485 |
| **Region** | | | | |
| North | 34.4 (33.7-35.1) | 33.5 (32.7-34.3) | 35.9 (34.7-37.1) | 27838 |

*(Continued)*

**Table 2.** (Continued)

| Variables | Proportion of women with undiagnosed hypertension (95% CI) | | | Total |
|---|---|---|---|---|
| | Total | Rural | Urban | |
| Central | 42.5 (41.8-43.1) | 42.7 (42-43.4) | 41.8 (40.3-43.3) | 29848 |
| East | 34.1 (33.2-34.9) | 34.8 (33.9-35.7) | 31.7 (29.8-33.6) | 20039 |
| Northeast | 37.5 (36.5-38.6) | 38 (36.8-39.1) | 35.9 (33.7-38.3) | 19857 |
| West | 52.5 (51-53.9) | 53.6 (52-55.2) | 51.1 (48.6-53.6) | 10328 |
| South | 48.1 (47.1-49) | 49.1 (48-50.3) | 46.7 (45.2-48.3) | 18654 |
| **Use of Internet** | | | | |
| Never | 43.8 (42.6-44.9) | 43.1 (41.9-44.4) | 45.6 (43-48.3) | 13847 |
| Yes | 38.4 (36.3-40.5) | 36.7 (34.3-39.2) | 39.9 (36.6-43.3) | 5144 |
| **Owns a mobile telephone** | | | | |
| Yes | 46.5 (45-48) | 45.6 (44-47.1) | 49.6 (46-53.2) | 8112 |
| No | 39.2 (37.8-40.6) | 38.2 (36.6-39.8) | 40.6 (38.1-43.1) | 10879 |
| **Usage of any type of tobacco** | | | | |
| No | 41.2 (40.8-41.6) | 40.9 (40.5-41.4) | 41.8 (41-42.6) | 116150 |
| Yes | 44.9 (43.4-46.3) | 46.1 (44.5-47.6) | 40.3 (36.6-44.2) | 10414 |
| **Current alcohol usage** | | | | |
| No | 41.3 (40.9-41.7) | 41.1 (40.6-41.5) | 41.7 (41-42.5) | 123097 |
| Less than once a week | 53.6 (49.4-57.7) | 55.3 (50.9-59.7) | 45.4 (33.5-57.9) | 1525 |
| Once a week | 57.4 (53.1-61.6) | 59.2 (54.8-63.5) | 45.5 (32.2-59.5) | 1413 |
| Everyday | 63.7 (57.5-69.4) | 65.2 (59.1-70.9) | 54.2 (34.2-73) | 529 |
| **Covered by Health Insurance** | | | | |
| No | 40.7 (40.3-41.2) | 40.1 (39.6-40.6) | 41.9 (41-42.8) | 85298 |
| Yes | 43.1 (42.4-43.7) | 43.8 (43.1-44.6) | 41.4 (40.1-42.8) | 41266 |
| **Type of health facility recently used** | | | | |
| None | 44.8 (44.3-45.3) | 45.1 (44.5-45.6) | 44.4 (43.4-45.4) | 78135 |
| Public facility | 37 (36.2-37.7) | 36.4 (35.5-37.2) | 38.5 (36.8-40.1) | 30984 |
| Private facility | 35.3 (34.3-36.3) | 34 (32.9-35.1) | 37.1 (35.3-39) | 17005 |
| Other | 33.5 (28-39.6) | 30.8 (24.7-37.7) | 39.8 (28.6-52.2) | 440 |
| **Anaemia** | | | | |
| Severe | 31.5 (29-34.1) | 30.9 (28.3-33.6) | 32.7 (27.4-38.5) | 2723 |
| Moderate | 38.4 (37.7-39.2) | 38.5 (37.7-39.3) | 38.2 (36.7-39.8) | 31898 |
| Mild | 40.3 (39.5-41.1) | 40.2 (39.4-41.1) | 40.5 (38.9-42.1) | 28685 |
| Not anaemic | 44.6 (44-45.1) | 44.1 (43.4-44.7) | 45.5 (44.3-46.6) | 57550 |
| **BMI** | | | | |
| Underweight | 39.6 (38.5-40.7) | 39.5 (38.3-40.6) | 40.1 (37.1-43.2) | 12735 |
| Normal weight | 42.4 (41.8-42.9) | 42.2 (41.6-42.8) | 42.7 (41.5-44) | 62736 |
| Overweight | 44.5 (43.7-45.3) | 44.5 (43.6-45.5) | 44.5 (43.1-45.9) | 31618 |
| Obese | 40.4 (39.3-41.6) | 40.3 (38.9-41.8) | 40.5 (38.7-42.3) | 13881 |
| **Waist to Hip Ratio** | | | | |
| Less or equal to 0.9 for men and 0.8 for femen | 41.2 (40.6-41.8) | 41.2 (40.6-41.9) | 41.1 (39.8-42.4) | 45257 |
| (Cut offs > 0.9 for men and >0.8 for women) | 42.1 (41.6-42.6) | 41.7 (41.2-42.3) | 42.8 (41.9-43.8) | 80156 |

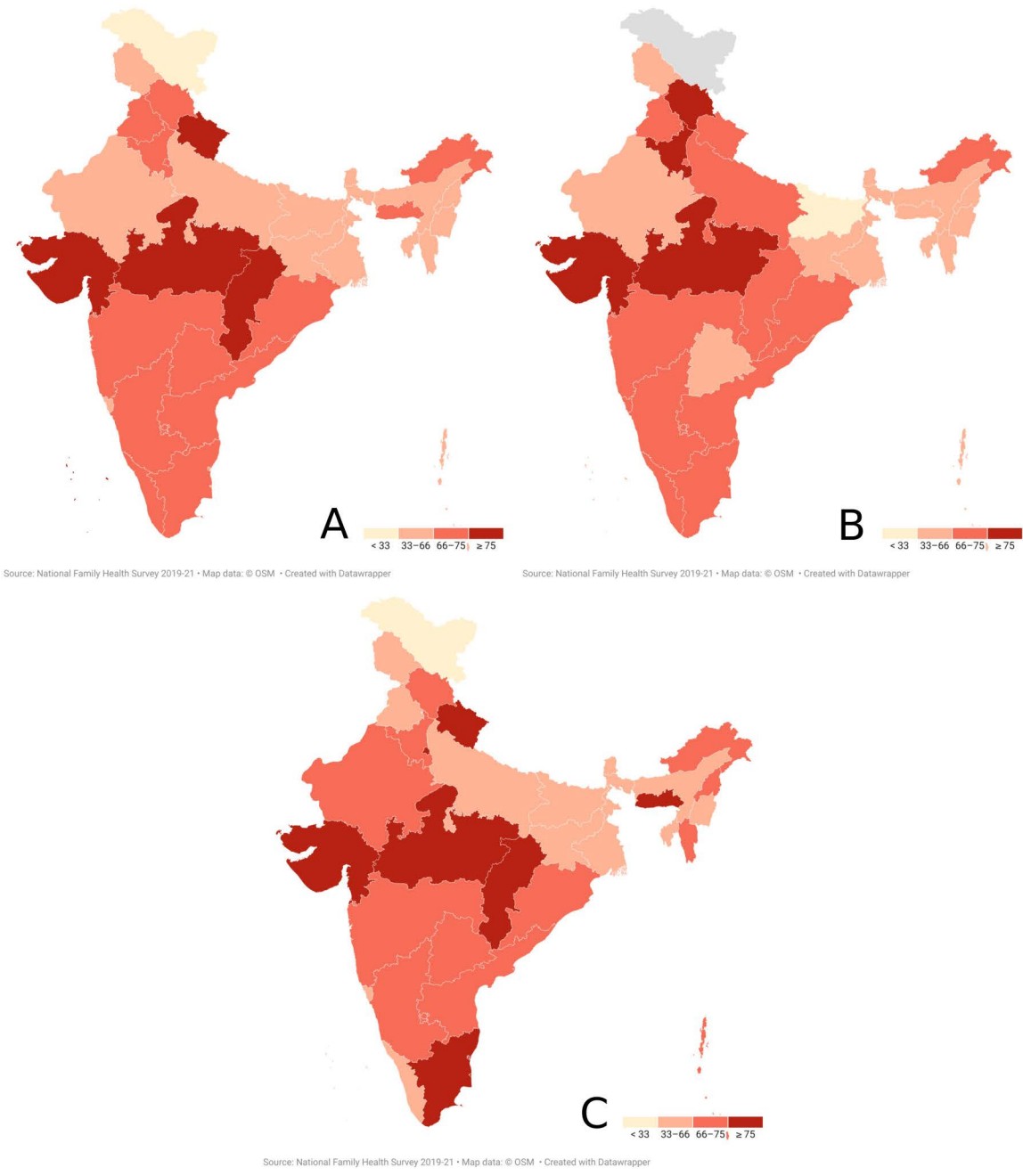

**Fig 2. State-wise proportional distribution of men with undiagnosed hypertension among all men with hypertension in the 15-49 year age group in India; A-Overall; B-Urban; C-Rural.**

like Chhattisgarh (Rural-67.7%; Urban-54.5%), NCT of Delhi (Rural-30.4%; Urban-46.2%), Sikkim (Rural-50.4%; Urban-33.5%), Nagaland (Rural-51.9%; Urban-33%), Mizoram (Rural-53.2%; Urban-39%) and Ladakh (Rural-23.5%; Urban-7.2%). States with the lowest proportion of women with undiagnosed hypertension are Ladakh, Bihar, Haryana, Himachal Pradesh and Punjab.

### Factors associated with undiagnosed hypertension

The proportion of men with undiagnosed hypertension (66.3%) was significantly higher than the proportion of women (41.4%) (p-value <0.001). Multivariable logistic regression analysis was done to calculate adjusted odds for factors associated with undiagnosed hypertension in men and women in the 15–49 year age group. Table 3 summarizes the protective factors and risk factors associated with undiagnosed hypertension among men and women across rural and urban areas.

Among men, age, usage of tobaaco products, recent usage of public or private health facility and health insurance coverage were found to be significant predictors for undiagnosed hypertension in urban areas. In rural areas, the important predictors were caste/tribe, religion, anaemia, working away from home, owning a mobile telephone and recent usage of public or private health facility. Among women, caste/tribe, marital status, wealth index, anaemia and recent usage of public or private health facility were found to be significant predictors for undiagnosed hypertension in urban areas. In rural areas, the important predictors were caste/tribe, daily alcohol usage, age, religion, higher education, wealth index, anaemia and recent usage of any health facility.

## Discussion

### Burden of undiagnosed hypertension

The prevalence of undiagnosed hypertension was found to be 11.7% among men and 7.2% among women. The proportion of undiagnosed hypertension among all those with hypertension was 66.3% among men and 41.4% among women. This means that a large section of the population with hypertension have remained undiagnosed in comparison to a proportion of less than 10% in countries with high treatment coverage [7]. Around 25–30% of women and men with hypertension in many sub-Saharan African and in some countries in south, and southeast Asia had uncontrolled hypertension who were not diagnosed or treated. Pooled analysis of several studies from India showed that around 20% men and women with uncontrolled hypertension had not been diagnosed or treated [7]. The differences between the estimates can be attributed to the difference in study definitions as well as the difference in study population, where the pooled analyses reported data for ages between 30 and 79 years. The findings indicate that it is possible to bring the proportion of undiagnosed hypertension far below current levels with expansion of screening, diagnostic and treatment coverage. It also reinforces the need to understand the factors behind undiagnosed hypertension so as to be able to bring more people under the umbrella of treatment and control.

Previous analysis of nationwide data on undiagnosed hypertension is scarce in India. An analysis of undiagnosed hypertension among women from NFHS-4 has since been retracted due to errors in classification [18]. A study on hypertension screening, awareness, treatment and control from the NFHS 4 dataset by Prenissl et al. found that 55.3% of those with hypertension were unaware of their status [6]. Analysis of the Longitudinal Ageing Study of India, Wave-1 (2017–18) which included adults above the age of 45 years, found the proportion of undiagnosed hypertension among all those with hypertension to be 42.3% (48.5% for men and 37.5% for women) [11]. The difference in findings can be explained by the different age groups of the respective study populations, but once again, it indicates that a sizeable proportion among even the elderly have remained undiagnosed.

### Rural-urban context in undiagnosed hypertension

Our study found an even distribution of undiagnosed hypertension between urban and rural areas in both men and women. This contrasts with the findings from the analysis of the LASI data [11], where prevalence of undiagnosed hypertension was 12.4% higher in rural areas as compared to urban areas. However, our state-level analysis found rural-urban differences in distribution of undiagnosed hypertension for both men and women with Delhi, Puducherry, Nagaland, Chhattisgarh, Bihar and Sikkim reporting around 20% higher prevalence of undiagnosed hypertension in rural areas.

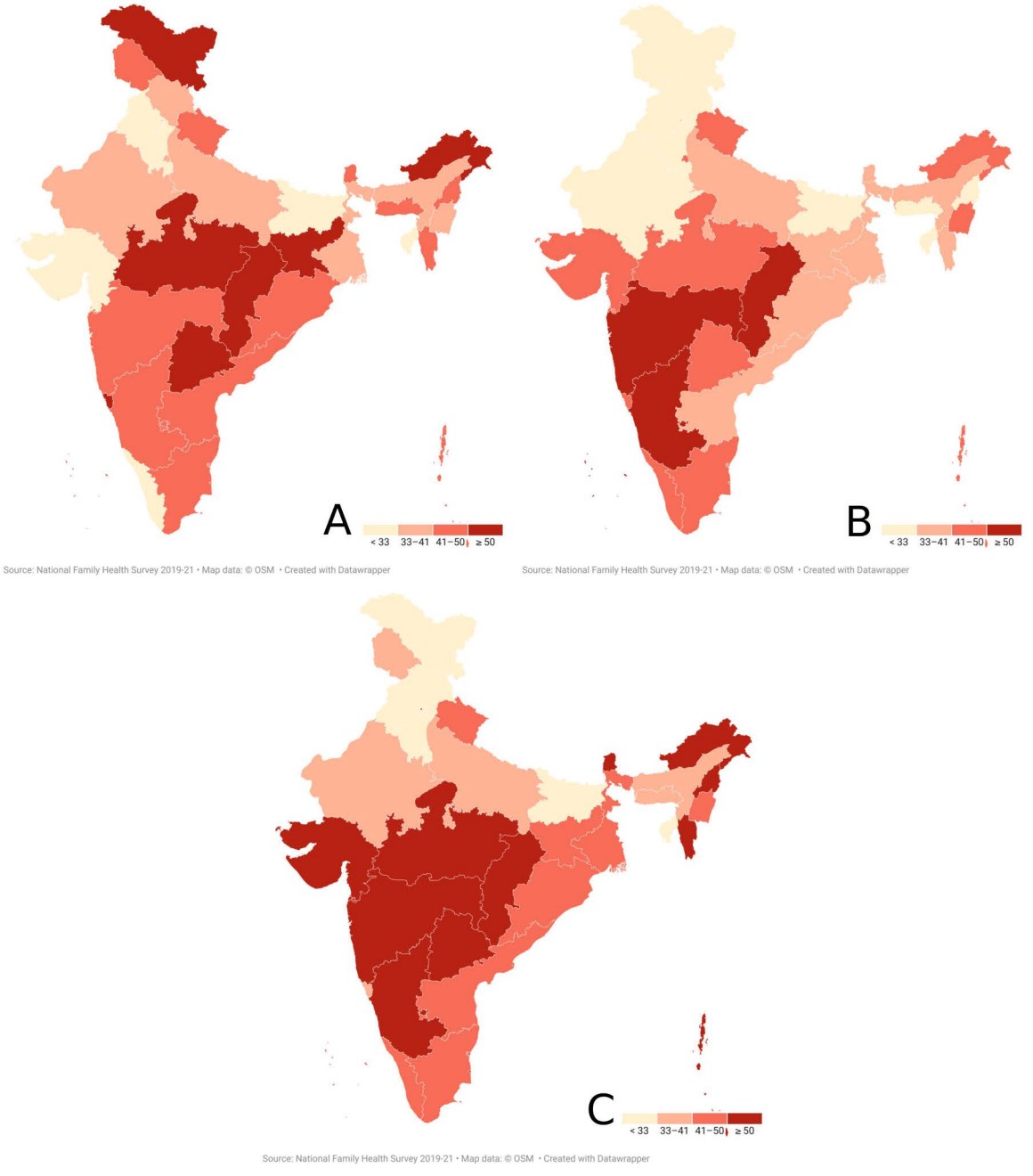

**Fig 3. State-wise proportional distribution of women with undiagnosed hypertension among all women with hypertension in the 15-49 year age group in India; A-Overall; B-Urban; C-Rural.**

The predictors for undiagnosed hypertension in urban and rural areas varied. Among men, ages 20–29 and 30–39 years were more likely to have undiagnosed hypertension in the urban areas. This may be an indication of the effect of the increasing adoption of unhealthy lifestyles and diet among the urban youth and middle-aged male population. This may also be due to the existing perception that hypertension is primarily a disease of the elderly, and the younger adults do not feel the need to be screened or tested. Conversely, rural women in the 15–19- and 20–29-year age groups were

**Table 3. Summary of risk factors and protective factors associated with undiagnosed hypertension among men and women aged 15-49 years from NFHS 5 (2019-21).**

| | | | Rural aOR (95% C.I.)** | Urban aOR (95% C.I.)** |
|---|---|---|---|---|
| **Men** | **Risk factors** | **Age** | | |
| | | 40-49 years | Ref | |
| | | 15-19 years | 0.78 (0.61-1.01) | 1.37 (0.88-2.13) |
| | | 20-29 years | 1.26 (1.11-1.45) | 1.43 (1.15-1.79)* |
| | | 30-39 years | 1.27 (1.15-1.4) | 1.39 (1.2-1.61)* |
| | | **Caste/Tribe** | | |
| | | Others | Ref | |
| | | Scheduled Tribe | 1.45 (1.25-1.68)* | 0.92 (0.7-1.21) |
| | | Scheduled Caste | 0.95 (0.83-1.09) | 1.02 (0.83-1.24) |
| | | OBC | 1.0 (0.89-1.12) | 1.12 (0.95-1.32) |
| | **Protective factors** | **Usage of any tobacco product** | 0.89 (0.79-1.01) | 0.77 (0.64-0.92)* |
| | | **Anemia** | | |
| | | Not anemic | Ref | |
| | | Mild | 0.86 (0.77-0.96) | 0.76 (0.63-0.92)* |
| | | Moderate | 0.74 (0.6-0.91)* | 0.65 (0.45-0.93)* |
| | | Severe | 0.39 (0.2-0.75)* | 0.56 (0.07-4.33) |
| | | **Religion** | | |
| | | Hindu | Ref | |
| | | Muslim | 0.61 (0.52-0.71)* | 0.97 (0.79-1.19) |
| | | Christian | 0.95 (0.79-1.14) | 1.02 (0.75-1.38) |
| | | Others | 0.83 (0.7-0.97) | 0.81 (0.6-1.09) |
| | | **Works away from home** | | |
| | | Works at home | Ref | |
| | | Works away from home | 0.75 (0.68-0.84)* | 0.87 (0.72-1.05) |
| | | **Owns a mobile telephone** | | |
| | | No | Ref | |
| | | Yes | 0.8 (0.69-0.94)* | 1.32 (0.92-1.88) |
| | | **Recent usage of health facility** | | |
| | | None | Ref | |
| | | Public | 0.68 (0.62-0.76)* | 0.74 (0.63-0.87)* |
| | | Private | 0.67 (0.58-0.76)* | 0.61 (0.5-0.73)* |
| | | Others | 0.87 (0.59-1.28) | 0.98 (0.41-2.35) |
| | | **Health insurance coverage** | | |
| | | No | Ref | |
| | | Yes | 1.1 (1.01-1.2) | 0.8 (0.71-0.93)* |
| **Women** | **Risk factors** | **Caste/Tribe** | | |
| | | Others | Ref | |
| | | Scheduled Tribe | 2 (1.91-2.1)* | 1.3 (1.17-1.42)* |
| | | Scheduled Caste | 1 (0.95-1.04) | 0.98 (0.92-1.06) |
| | | OBC | 1.15 (1.11-1.19) | 1.11 (1.05-1.17) |
| | | **Current alcohol usage** | | |
| | | None | Ref | |
| | | Less than once a week | 1.19 (1.05-1.35) | 1.02 (0.78-1.32) |
| | | Once a week | 1.21 (1.07-1.37) | 1.08 (0.76-1.53) |
| | | Everyday | 1.39 (1.13-1.7)* | 1.46 (0.8-2.65) |

*(Continued)*

**Table 3.** (Continued)

|  |  |  | Rural<br>aOR (95% C.I.)** | Urban<br>aOR (95% C.I.)** |
|---|---|---|---|---|
| **Protective factors** | **Age** |  |  |  |
|  | 40-49 years | | Ref | |
|  | 15-19 years | | 0.64 (0.59-0.7)* | 0.93 (0.8-1.08) |
|  | 20-29 years | | 0.64 (0.61-0.67)* | 0.83 (0.77-0.9) |
|  | 30-39 years | | 0.91 (0.88-0.94) | 1.06 (1-1.12) |
|  | **Marital Status** | | | |
|  | Never in union | | Ref | |
|  | Currently married | | 0.6 (0.56-0.64)* | 0.68 (0.62-0.76)* |
|  | Widowed/Separated/Divorced | | 0.58 (0.53-0.63)* | 0.67 (0.59-0.77)* |
|  | **Religion** | | | |
|  | Hindu | | Ref | |
|  | Muslim | | 0.76 (0.72-0.8)* | 0.93 (0.86-0.99) |
|  | Christian | | 0.8 (0.79-0.9)* | 0.81(0.72-0.91) |
|  | Others | | 0.82 (0.77-0.87) | 0.81 (0.73-0.91) |
|  | **Education** | | | |
|  | No education/Pre-primary education | | Ref | |
|  | Primary education | | 0.89 (0.85-0.93) | 0.98 (0.9-1.07) |
|  | Secondary education | | 0.82 (0.79-0.85) | 0.88 (0.82-0.94) |
|  | Higher education | | 0.76 (0.71-0.81)* | 0.85 (0.78-0.93) |
|  | **Wealth Index Quintile** | | | |
|  | Poorest | | Ref | |
|  | Poorer | | 0.87 (0.84-0.91) | 0.87 (0.74-1.03) |
|  | Middle | | 0.83 (0.79-0.87) | 0.81 (0.69-0.95) |
|  | Richer | | 0.79 (0.75-0.83)* | 0.77 (0.66-0.9)* |
|  | Richest | | 0.75 (0.71-0.8)* | 0.73 (0.63-0.86)* |
|  | **Anemia** | | | |
|  | Not anemic | | Ref | |
|  | Mild | | 0.84 (0.81-0.87) | 0.82 (0.77-0.87) |
|  | Moderate | | 0.78 (0.75-0.81)* | 0.76 (0.72-0.8)* |
|  | Severe | | 0.54 (0.49-0.6)* | 0.5 (0.42-0.59)* |
|  | **Recent usage of health facility** | | | |
|  | None | | Ref | |
|  | Public | | 0.72 (0.7-0.75)* | 0.78 (0.73-0.83)* |
|  | Private | | 0.61 (0.59-0.64)* | 0.69 (0.65-0.74)* |
|  | Others | | 0.56 (0.43-0.71)* | 0.7 (0.47-1.05) |

*Factors of relevant public health significance

**Adjusted for all socioeconomic, anthropometric and health-related beheviour variables

less likely to have undiagnosed hypertension. The fact that all pregnant women receiving antenatal care are screened for hypertension may have contributed to the higher proportions of known hypertension in these age groups. Though, community based and opportunistic screening for hypertension has been implemented across the country through the National Programme for Prevention & Control Of Cancer, Diabetes, Cardiovascular Diseases & Stroke (NPCDCS) for those above the age of 30 years [19], the high proportions, especially in the 30–39 year age groups indicate that the coverage is

nowhere near the necessary levels even within the urban healthcare system. Scheduled tribes have consistently higher odds of having undiagnosed hypertension in the rural areas among both men and women. This reinforces previous policy priorities and highlights the need to continue to treat tribal health as a priority if we are to achieve universal health coverage [20]. Having ever been married was associated with lower odds of undiagnosed hypertension among both urban and rural women. Similar findings have been reported from previous studies in India [6,11,21]. This may indicate better health seeking behaviour among ever married women as compared to those who had never been in a union. Higher educational attainment was associated with lower odds of undiagnosed hypertension among women. The association was of public health significance in rural women. There has been previous evidence for the same [6,11,21]. The findings reiterate the importance of education among women and it's likely impact on their health-seeking behaviour, especially in the rural areas. Higher wealth index was associated with lower odds of having undiagnosed hypertension among women in both urban and rural areas. Similar association was not observed among men. This may indicate a link between socioeconomic status and women empowerment, which also ties in with health-seeking behaviour [22].

Being diagnosed with moderate or severe anaemia was associated with lower odds of undiagnosed hypertension among men and women in both rural and urban areas, which may be due to their increased contact with the health system for associated treatment. The effectiveness of opportunistic screening for hypertension has been well documented. [23,24] Usage of tobacco is associated with lower odds of undiagnosed hypertension among urban men. This may be due to higher contact of the tobacco users with the health system due to other health issues, which may have led to opportunistic screening and diagnosis of hypertension. This also indicates better healthcare seeking behaviour among this population – as compared to the rural men where the consumption of tobacco is less often associated with health risks and social taboos [25]. Current daily alcohol usage was associated with higher odds of undiagnosed hypertension among rural women. Poor health-seeking behaviour has been previously reported among alcohol users and this may explain our finding [23,24].

Working away from home was associated with lesser odds of having undiagnosed hypertension among rural men. While occupation did not show an association of public health significance, employment was significantly associated with lower odds of undiagnosed hypertension (S1 Table). This may be because of the existing occupational health initiatives by public and private entities to ensure provision of preventive, promotive and curative care for the employed [26]. Owning a mobile telephone was also associated with lesser odds of having undiagnosed hypertension among rural men. Having access to mobile phones has been previously known to have a positive effect on disease awareness and health-seeking behaviour [27]. Having health insurance coverage was associated with lower odds of having undiagnosed hypertension among urban men. Literature indicates that having health insurance coverage encourages increased medical check-ups [28]. However, the pattern is not seen in rural areas and among the poor [29], which needs to be further explored. Visit to public or private health facility was significantly associated with lower odds of undiagnosed hypertension among both men and women in both urban and rural areas. While this indicates a good performance of the facility-based screening programme, it calls into question the effectiveness of community-based risk assessment and screening for non-communicable diseases.

## Strengths and limitations

To the best of our knowledge, this is the first study to analyse the NFHS-5 dataset for factors associated with undiagnosed hypertension and their distribution pattern across rural and urban areas. The large sample size and the uniform methodology of the nationwide survey makes the results reliable and generalizable. State-level analysis of the data along with an urban-rural breakdown allowed for the highlighting of high priority areas where interventions may need to be stepped up or alternative approaches may need to be considered.

This study has various limitations. Firstly, it is based upon secondary data available from a cross-sectional survey that assesses relationships at one point in time. This also means that causation cannot be inferred and only possible

associations can be discussed. Secondly, there is a possibility of recall bias and low truth quotient associated with all survey instruments. Thirdly, application of American Heart Association thresholds of 130/80 for stage 1 hypertension, would have substantially increased the number of people with undiagnosed hypertension. However, since the cut-off of 140/80 mm Hg is used for diagnosing and treating hypertension in India, this cut-off was used in this study. Fourthly, the survey population was restricted to men and women aged 15–49 years and this means that we have not been able to account for a major proportion of those affected, i.e., men and women over 50 years of age. Fifthly, NFHS-5 was conducted in two phases, one before the COVID-19 pandemic, and one during the pandemic. It is not possible to estimate how the disruption of regular services may have affected the burden of undiagnosed hypertension. While many who were undiagnosed, might have come in contact with the health system due to the pandemic and gotten diagnosed, others may have remained undiagnosed due to disruption of regular screening and diagnostic services at both urban and rural areas. In addition, factors such as dietary sodium intake, physical activity, psychosocial stress, sleep patterns, and family history of hypertension, which are known to be associated with both hypertension risk and health-seeking behavior, were not captured in the NFHS-5 dataset. Access-related variables such as distance to health facility or quality of services were also not part of the survey. Future research should incorporate these variables through primary data collection or augmented survey instruments to better understand the multifactorial nature of undiagnosed hypertension and improve causal inference. Finally, while choropleth maps were used to visually depict state-wise differences in the proportion of undiagnosed hypertension, this was not further assessed using statistical tests. Given the complex survey design and weighted data, a Rao–Scott adjusted chi-square test or multilevel modelling may be incorporated into future analyses to validate the significance of geographic variability and account for clustering and stratification in the survey design.

## Implications for policy and practice

Improvement of health seeking behaviour among women may be brought about through education and empowerment by means of livelihood and increased earning potential. Special focus may be needed to improve health seeking behaviour of those who had never been married. Health-seeking behaviour among current alcohol users needs to be improved, through de-stigmatization of the topic, especially among women and through easy availability of cessation services. Screening of all those who report current alcohol or tobacco usage for hypertension may be incorporated as part of the national programme guidelines. High quality screening, diagnostic and treatment services for hypertension needs to be made accessible at all healthcare facilities at primary, secondary and tertiary levels. Community-based screening and referral under NPCDCS needs to be strengthened to reduce the burden of undiagnosed hypertension, especially in the rural areas of States identified with a high rural-urban gap in prevalence. Health promotion through the platform of mobile messaging and social media may be one the ways forward to draw the attention of the rural and migrant youth towards adoption of healthy lifestyles and the importance of early screening and diagnosis of hypertension. Improvement of access to health insurance coverage among the rural population is also likely to reduce the burden of undiagnosed hypertension.

## Future research

Health technology assessment studies to explore the possibilities of changes to the NPCDCS guidelines for screening, diagnosis and treatment of hypertension, may be considered to expand the coverage of hypertension awareness and treatment, with focus on the urban youth. The findings of this analysis open avenues for further research on interventions to improve health-seeking behaviour in the rural areas, as well as among current alcohol users. Predictors such as health insurance coverage, working away from home and owning a mobile telephone needs to be explored as possible areas of intervention to further reduce the burden of undiagnosed hypertension in the future. Potential improvement in quality of life, increase in life expectancy and reduction in DALYs due to undiagnosed hypertension may be estimated to better understand the benefits that would be accrued as a result of reducing the burden of undiagnosed hypertension.

## Conclusion

Our analysis of nationally representative data from the NFHS 5 (2019–21) revealed that two-thirds of men and two-fifths of women with hypertension were previously undiagnosed. State-level analysis highlighted multiple states with higher proportions of undiagnosed hypertension, both in urban and rural areas. Guidelines for population-based screening of common NCDs including hypertension has been implemented through the NPCDCS under the National Health Mission.[26] It is also operationalised through the Ayushman Bharat Health and Wellness Centres as part of the comprehensive primary health care package [30]. The findings of this study show the ground that needs to be covered in terms of diagnosing all who have hypertension. Only when one is aware of their status, will it be possible to bring them under treatment and ultimately ensure effective control of disease. Several predictors identified in this study such as age, caste, education, wealth index, alcohol and tobacco use, health insurance coverage, working away from home and owning a mobile telephone, all provide interesting avenues for further research, in order to inform necessary change in policy and practices in the future. The problem of undiagnosed hypertension is not restricted to only the rural areas but has important predictors in urban areas as well, which need to be explored further and acted upon. There exists an urgent need to plan interventions with an understanding of the health seeking behaviour of different sub-populations. Strengthening and expansion of community-based screening for hypertension appears to be the most important intervention to be able to identify apparently healthy individuals who are unlikely to visit the health facilities.

## Supporting information

**S1 Table. Factors associated with undiagnosed hypertension (adjusted odds ratio with 95% confidence intervals) among men and women aged 15-49, India, NFHS-5, 2019–21.**
(DOCX)

## Acknowledgments

The authors would like to acknowledge the contribution and mentoring received from Dr. Hemant D Shewade, Scientist E (Medical), Division of Health Systems Research, ICMR-National Institute of Epidemiology, Chennai, Tamil Nadu, India for this research proposal and manuscript writing. The authors would also like to thank the team of Resource Center for Tobacco Control, Department of Community Medicine and School of Public Health, PGIMER, Chandigarh.

## Author contributions

**Conceptualization:** Aritrik Das, Yukti Bhandari, Jugal Kishore, Sonu Goel.

**Data curation:** Yukti Bhandari, Angad Singh.

**Formal analysis:** Aritrik Das, Angad Singh.

**Investigation:** Aritrik Das.

**Methodology:** Aritrik Das, Yukti Bhandari, Angad Singh.

**Project administration:** Jugal Kishore.

**Software:** Angad Singh.

**Supervision:** Jugal Kishore, Sonu Goel.

**Validation:** Angad Singh, Sonu Goel.

**Visualization:** Aritrik Das, Angad Singh.

**Writing – original draft:** Aritrik Das, Yukti Bhandari, Jugal Kishore.

**Writing – review & editing:** Aritrik Das, Yukti Bhandari, Jugal Kishore, Sonu Goel.

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
